# Four Questions in Cellular Material Design

**DOI:** 10.3390/ma12071060

**Published:** 2019-03-31

**Authors:** Dhruv Bhate

**Affiliations:** The Polytechnic School, Ira A. Fulton Schools of Engineering, Arizona State University, Mesa, AZ 85212-6300, USA; dhruv.bhate@asu.edu; Tel.: +1-480-727-1332

**Keywords:** cellular materials, honeycomb, lattice, foam, relative density, additive manufacturing, design

## Abstract

The design of cellular materials has recently undergone a paradigm shift, enabled by developments in Additive Manufacturing and design software. No longer do cellular materials have to be limited to traditional shapes such as honeycomb panels or stochastic foams. With this increase in design freedom comes a significant increase in optionality, which can be overwhelming to the designer. This paper aims to provide a framework for thinking about the four key questions in cellular material design: how to select a unit cell, how to vary cell size spatially, what the optimal parameters are, and finally, how best to integrate a cellular material within the structure at large. These questions are posed with the intent of stimulating further research that can address them individually, as well as integrate them in a systematic methodology for cellular material design. Different state-of-the-art solution approaches are also presented in order to provoke further investigation by the reader.

## 1. Introduction

Additive Manufacturing (AM) has enabled a wide exploration of the design space for engineering applications. One specific aspect within this design space is the development of cellular materials. Cellular materials have been part of purposeful engineering practice for several decades, and are well summarized in books [1,2,3]. They have found wide adoption in an extensive range of applications such as sandwich paneling, packaging, acoustic lining, catalytic converters, and heat exchangers. AM processes are enabling new design capabilities that were hitherto not accessible. Prior to the widespread adoption of AM, cellular materials were typically limited to honeycomb sandwich panels and stochastic foams. With AM, it is now possible to architect cellular materials that are limited only by the resolution of the AM process in question. With this additional freedom to the designer, there comes a greater responsibility of ensuring optimality of selection among several choices. When it comes to incorporating cellular material designs into structures, there are four main questions the designer needs to address, and these are the focus of this paper:What is (are) the optimum unit cell(s)?How should the size of the cells vary spatially?What are the optimal cell parameters?How best should the cells be integrated with the larger form?

In the following discussion, each of these questions is examined in turn and the range of options available is demonstrated. The methods of selecting between options, ranging from analytical, experimental, computational, and biomimetic, are briefly discussed for each question, with references included to guide readers to seek these answers out in contexts relevant to them. Towards the end of this paper, key challenges in addressing these questions are identified, with the hope of promoting further research in these areas in the future, and developing a comprehensive methodology in cellular material design.

## 2. Question 1: What is the Optimum Unit Cell?

A designer can select from, quite literally, an infinitely large list of unit cells, ranging from the prismatic hexagonal honeycomb to stochastic foams. The first question therefore deals with how one should go about making this selection for a specific application. If one is to revert to first principles, as opposed to replicate existing designs, one way of approaching this question is to first classify this wide range of choice into three specific levels of decision making, as shown in Figure 1 [4]: Tessellation (the division of space into smaller, repeating entities), Elements (the material constituents of that space), and Connectivity (the ways in which these constituents are connected). 

Tessellations can either be periodic, stochastic or hierarchical, as proposed in reference [4]. Elements are either beam based, such as what we commonly associate with lattice structures [5,6,7], or surface based, such as in Triply Periodic Minimal Surfaces (TPMS) [8]. At a practical level, these three decisions can be used to realize a range of cell shapes, some examples of which are shown in Figure 2, and which is inspired by the design process in the nTopology Element lattice design software [9]. In addition to guiding a designer towards selecting a specific unit cell, a classification such as the one proposed here enables a deeper investigation of the relative benefits of each decision point (marked as a red dot in Figure 1). For example, one such question of deep significance is: under what circumstances should a periodic cellular material be preferred over a stochastic one?

Among the four questions, the selection of a specific unit cell has received the most consideration in the literature, and there are several methods to address these, reviewed in reference [4] and summarized here only briefly. These methods can be analytical such as the Maxwell’s stability criterion [10], modified for cellular materials [11], or derive from selection approaches based on beam theory such as the “Gibson–Ashby” models of cellular materials [1]. The former approach enables a classification of a cellular shape as being either stretch- or bending-dominated, wherefrom further conclusions may be drawn about the applicability of that shape for a specific desired function (high energy absorption or high stiffness, for example). The Gibson–Ashby modeling approach allows for a more detailed relationship between material properties and geometric parameters. 

Empirical methods, either computational or experimental, are also used to select among different cell shapes, and also used to determine how closely these shapes approach theoretical bounds such as Hashin–Shtrikman bounds [12]. The vast majority of literature concerning the behavior of AM cellular materials involves some empirical evaluations of cellular material response. Biomimicry has also been proposed as a way to help guide the selection of unit cell shapes depending on the loading conditions [13]. Most commercially available design software leave the selection of the unit cell shape to the designer, some software companies are working to integrate this selection through a “Material Designer” module that helps the designer use simulation to compare among geometries, extending capabilities previously developed for the general class of composite materials [14]. 

## 3. Question 2: How Should the Size of the Cells Vary Spatially?

Once a cell shape is selected, the larger structure needs to be populated with these cells. The main concern then is one of cell size, and its distribution—and the question for the designer essentially becomes one of selecting the best distribution for a specific structure, in a given application. With cellular materials, a designer has the option of defining a regular periodic structure, or a stochastic structure with some governing sizing rule—both these examples are shown in Figure 3a,b, respectively.

The selection of the size of cells is bounded at the low end by manufacturing constraints such as resolution and the need for powder removal (in the case of powder based AM processes). The maximum overhang distance may, for some processes, define an upper bound. Between these bounds imposed by manufacturing processes however, it is not obvious what specific size a cellular material should have, or how it should vary spatially. Crucial to enabling this selection is the concept of relative density [1], which can, as is more commonly the case, also be modified through a tuning of parameters such as thickness of the members constituting the unit cell, which is discussed in the next section. 

Most commercial design software leaves the prescription of cell size and its distribution to the user and performs optimization at the level of the cell parameters [9]. Published literature on topology optimization based methods for lattice generation also tends to focus on parameters, and not cell size optimization [15,16]. In work done by Cheng et al., the discretized cell size is selected to conform to the geometry of the component being filled with cellular materials [15]. An alternative method of specifying cell size was developed by Brackett et al. [17], using error diffusion to generate dithered points that are then used as seeds for Voronoi (or related Delaunay) cells. 

Finally, a biomimetic approach may also help address the question of cell size distribution. Gradients are common in many natural cellular materials, two examples of which are shown in Figure 4a,b, and can be implemented in design software, as shown in Figure 4c. The specific property that varies can take many forms: in a review paper, Liu et al. [18] classify biological gradients into six categories: composition, arrangement, distribution, orientation, dimension, and interface. Most appropriate for the discussion here, the “dimension” category refers to gradients in materials through “alterations in their characteristic structural dimensions”. The cited benefits of gradients from a structural standpoint include the modulation of hardness, stiffness, and toughness. Another case of cell size control is seen in insect wings and in leaf patterns, where cells emerge from an overarching branching pattern. The branches themselves often have a non-structural role such as enabling fluid flow, but additionally play a structural role by stiffening the entity [19]. 

## 4. Question 3: What Are the Optimal Cell Parameters?

The previous two questions dealt with selection of a unit cell topology and its size distribution through a structure of interest. Nothing has yet been said about the dimensions and material composition of the cell itself. For geometric parameters, consider the relatively straightforward case of the hexagonal honeycomb: a typical way to represent this structure parametrically is to describe it in terms of the length (*l*) and thickness (*t*) of the walls and the corner radius (*r*), as indicated in Figure 5. While length is implicitly selected based on the unit cell shape and size selections in the previous two questions, thickness and radii can be prescribed independently. 

Beyond length, thickness, and junction parameters, the cross-section of the connecting element itself can vary. In the case of beam-based lattice structures for example, the cross-section can assume a range of shapes, including a variable section from one end to another. Figure 6 shows three possible cross-sections one may assign to a beam. Depending on the scales involved relative to the manufacturing process of interest, such nuances may not be resolvable, but if they are, they become a design variable that can influence behavior as well as manufacturability. For example, the teardrop shape in Figure 6c, is so designed as to enable overhanging lattice beams to be self-supporting for processes that need support, such as in powder based AM processes. Finally, the properties of the constituent material that make up the cellular structure should also be considered as a parameter since they are subject to selection. Of interest to the designer of cellular materials is then to understand how best to optimize these parameters to attain certain performance objectives.

From an analytical standpoint, scaling laws based on relative density are a commonly used method to prescribe thickness parameters. Relative density has been used as a critical figure-of-merit in the industrial application of foams [2]. It can be used in conjunction with property equations to make design decisions when density is of importance—in buoyancy or light-weighting applications, for example. The relative density is expressed as *ρ**/*ρ_s_*, where *ρ** is the density of the cellular material, and *ρ_s_* the density of the material of which the cellular structures are made. The relative density can then be calculated from the geometry of the shape, and for beam-based structures, it is typically some function of the ratio of the thickness of the member (edge or wall) to its length (*t*/*l*). For honeycombs and foams, for example, these relationships typically take the following forms, where *C*_1_, *C*_2_, and *C*_3_ are constants [1]:(1)$$\mathrm{Honeycomb}:\;\frac{\rho^{*}}{\rho_{s}}=C_{1}\frac{t}{l}$$
(2)$$\mathrm{Open}\;\mathrm{Cell}\;\mathrm{Foam}:\;\;\frac{\rho^{*}}{\rho_{s}}=C_{2}{\left(\frac{t}{l}\right)}^{2}$$
(3)$$\mathrm{Closed}\;\mathrm{Cell}\;\mathrm{Foam}:\;\frac{\rho^{*}}{\rho_{s}}=C_{3}{\left(\frac{t}{l}\right)}^{3}$$

A table of such relationships for different shapes can be found in reference [1]. For more complex shapes, relative density can be computed from an output of the design file used to create it, and it can also be measured experimentally. However, a designer is concerned with selecting a certain shape and its associated parameters, not in knowing its relative density per se. Relationships between a certain property one is looking to design with, and the relative density, commonly termed as “Scaling Laws”, are of help in this regard. These scaling laws can be developed for mechanical, thermal, electric, acoustic, and other areas of interest, either analytically or empirically through experimentation or simulation.

Scaling laws are well-developed for making predictions on mechanical response. For the effective modulus, for example, the relationship typically takes the following form, where *E** and *E_s_* are the effective modulus of the cellular material and of the bulk solid, respectively [1]:(4)$$E^{*}=C\;E_{s}\;{\left(\frac{\rho^{*}}{\rho_{s}}\right)}^{n}$$

In principle, scaling laws can be developed for any property where physical principles suggest the existence of a relationship between that property and the relative density of the cellular material. In practice, these laws are developed by manufacturing and testing materials at a range of relative density values and then fitting an appropriate relationship to the observed datasets. This method, while empirical and geometry-dependent, can prove to be a useful design protocol in the absence of analytically derived relationships.

It is important to remember that scaling laws do not explicitly address geometric detail—thus whether the relative density is obtained using variation of parameters (changing thickness of walls and struts, for example), or through changing cell sizes, as shown in Figure 3a, or a combination of both strategies, are all equivalent for the purposes of property estimation. For the effective properties discussed here, these may be reasonable strategies. For studying the failure of cellular materials, they are likely to be inadequate. Gibson and Ashby [1] developed models based on beam theory that are a more robust representation of the relationships between parameters and mechanical behavior, but these models tend to be restricted to simpler geometries such as honeycombs and foams with specific geometries. Further, spatial variation of these parameters based on the analytical solutions in a structure with a heterogeneous state of stress, is non-trivial, and requires a computational approach.

Cellular material parameter optimization within a structure can be thought of as an extension of topology optimization concepts. There are at least three ways to implement this—the first involves discretizing a given component geometry into tessellated elements, within each of which a local topology optimization is performed. One such method that employs this approach is the Lattice Structure Topology Optimization (LSTO) method proposed by To and his co-workers [15]. Similar approaches have also been previously developed by Sigmund and others [20,21]. An alternative approach is to overlay cellular structures onto a structure already topologically optimized. This is at the heart of the method developed by Panesar et al. [16], and similar ideas have been implemented in commercial software.

Some software, such as nTopology Element (New York, NY, USA), permits modulation of thickness parameters in three different ways: (1) as a stipulated value applied globally, or (2) one varying spatially per a prescribed function (modifier), or (3) the solver locally optimizes the thickness of cells in response to a global load case (see Figure 7a–c, respectively). 

In addition to geometry alone, natural structures co-optimize both geometry and material composition in interesting ways, leveraging readily available materials [22]. One such example is the range of social insect nests found in nature, which can be composed of beeswax, mud or paper, as shown in Figure 8 [23]. A useful line of investigation is to study how combined design of material and structure for cellular materials can result in a more resource neutral design process, especially of interest for places where in-situ resource utilization is of interest such as space exploration and remote area construction. Another opportunity for bio-inspired exploration is the examination of subtler features in natural cellular materials such as the corner radii at junctions, and the variation in beam cross-sections, examples of which are shown in Figure 9. The role of corner geometry in cellular material has not received much study, but has been shown to significantly influence behavior [24]. From an engineering perspective, it is well-known that fillets can mitigate stress concentrations. Researchers have also shown that the corner radius plays a significant role in manipulating the stiffness behavior of honeycomb structures [25].

## 5. Question 4: How Best Should the Cells Be Integrated with the Larger Form?

The previous three design elements concerned the definition of a cellular pattern. A crucial question that remains is regarding how best one should integrate this pattern into a form of interest. More specifically, one may ask: what are the best ways to terminate cellular materials at an outer boundary?

The most commonly available method of introducing a cellular material into a solid is to use uniform infilling, shown in Figure 10. When a uniform infill is used, the assumption most often made is that the cellular material completely replaces the larger component with no remnant skin, or that the cells will be trimmed by a skin, which can result in dangling beams or unsupported walls. The termination of a uniform infill can be specified in at least three ways: by defining the termination either such that all cells lie completely inside the structure, or just the centroids do, or in a way that completely fills the structure with cells (Figure 10b–d, respectively). Uniform infilling is a useful approach to designing specimens for characterization, and may be a beneficial strategy when working with surface based cellular materials, but generally speaking is not preferable for components since exposed struts or walls may be prone to damage and infiltration. 

To avoid the risk of having struts protrude outside the geometry, conformal strategies have been developed by researchers [26,27] and also implemented in some commercial software, an example of which is shown in Figure 11a. In conformal geometries, the cellular material is wrapped in such a way as to ensure the outermost unit cells terminate exactly on the walls of the specified structure. Wrapping is now a feature in some cellular material design software, and can extend to complex geometries as shown in Figure 11b, but tends to distort the cellular geometry close to the skin boundaries. 

## 6. Discussion: Research Opportunities

Of the four questions discussed here, only the third (parameter optimization) is consistently addressed in commercial software and in most of the literature. However, it is not always apparent what the best cellular material shape or size for a particular application is (Questions 1 and 2), nor how best to integrate it into a 3-dimensional structure (Question 4) with heterogeneous states of stress. This is particularly true when the application demands multi-functionality, such as requiring a tailored stiffness at different locations, specifying the location of the center of gravity, and allowing for channels that enable thermal management—as shown in the case of the wing concept in Figure 12, adapted from reference [6].

A key challenge with unit cell selection is that it is often performed based on inferences drawn from studies conducted on unit cell, or finite domain cellular structures, which do not mimic the real structure and its complex state of stress. A unit cell that performs well under compression, for example, may not be ideal as the cell of choice for a torsional structure. This challenge becomes even more complex when multi-physics is involved, optimizing for compressive stiffness and thermal expansion, for example. A key research area is the development of tools that can rapidly evolve the unit cell shape in response to the local state, which in turn is driven by global boundary conditions, loads and objectives, and solve the “inverse homogenization” problem. 

There are several challenges with implementing cell size distribution algorithms in cellular material design software. For all cell shapes, the connections between cell members need to be enforced to avoid discontinuities, which can be problematic if the cell shapes are non-stochastic and vary spatially. The relative contributions of cell shape selection, size distribution, and parameter optimization to the performance of a structure is not at present a well-understood problem. It is not clear, for example, that optimizing cell size brings with it any inherent advantages over optimization of the parameter space for the cells themselves, such as local thickening of members where stresses are high. 

Finally, the integration of cellular materials into 3D structures is an area that would benefit from more research. Termination strategies have not been studied in great detail, especially in the context of manufactured parts and their failure signatures under complex loading conditions or multi-physics environments. Integration of cellular materials in levels of hierarchy also is an exciting research opportunity—consider the mimicking of insect wings, where the veins provide both flow and stiffening functions.

The design community is at the beginning of a new era in cellular material design, enabled by additive manufacturing and powerful computational tools. This era will see designers working with unprecedented freedom to design, and with this freedom comes the responsibility of asking the right questions and addressing them through design. In that sense, good design represents the manifestation of a designer’s understanding of how the world works, whether it is arrived at through tinkering and experimentation, or through model-driven approaches. With cellular materials, designers will need to combine both, to ensure cellular materials reach their full potential as enablers of a more sustainable, and interesting, world. 

## Figures and Tables

**Figure 1 materials-12-01060-f001:**
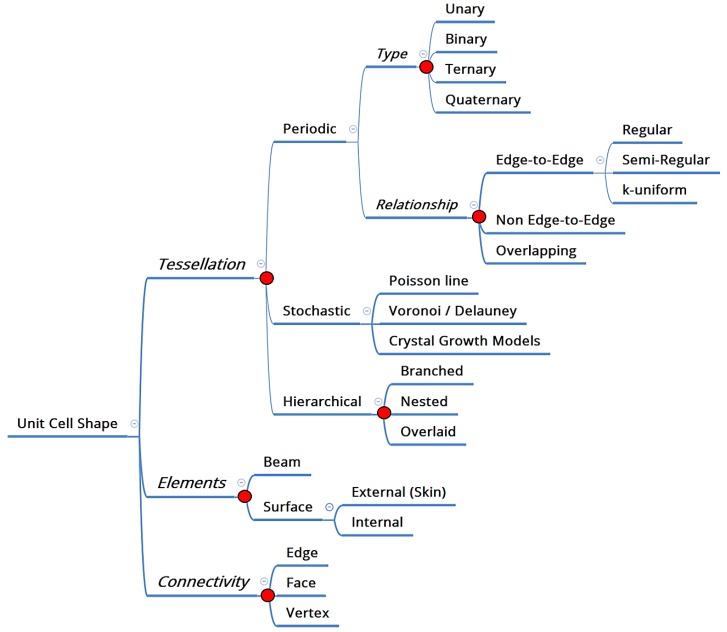
Classification of cellular materials unit cell design based on a three level decision-making process first proposed in reference [4].

**Figure 2 materials-12-01060-f002:**
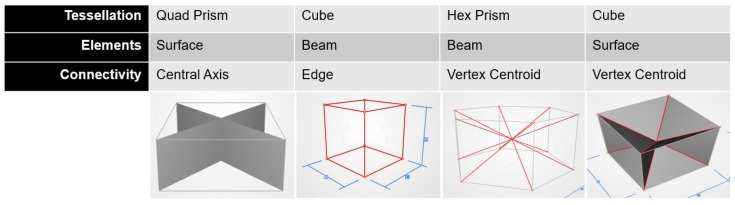
Four examples of how the three design decisions in Figure 1 translate into the realization of cellular shape designs.

**Figure 3 materials-12-01060-f003:**
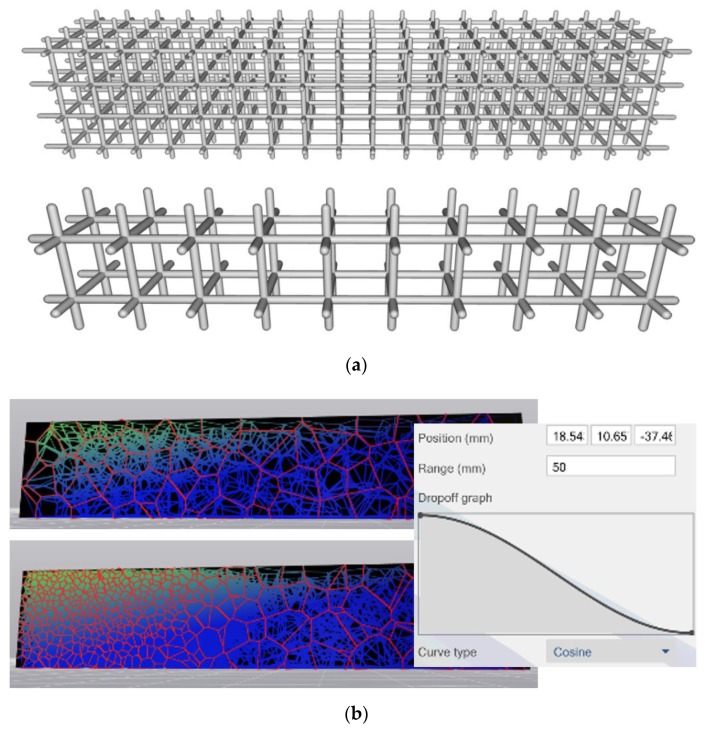
Cells can be distributed in a prescriptive manner, (**a**) shown for a periodic lattice, or (**b**) by using a function, shown for a stochastic lattice.

**Figure 4 materials-12-01060-f004:**
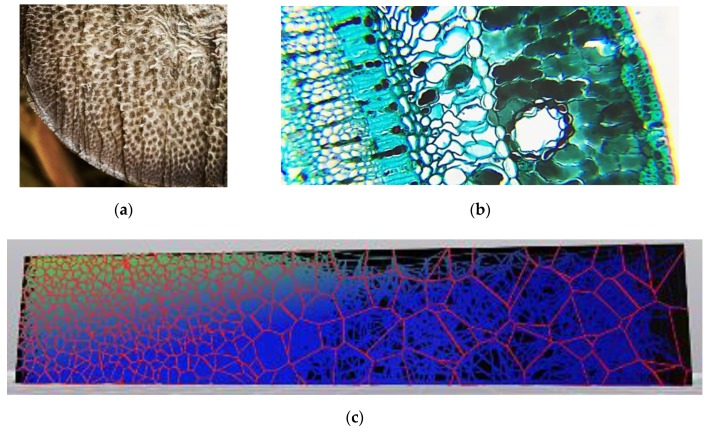
Gradients occur commonly in natural cellular materials, as shown in (**a**) sections of bamboo, (**b**) a pine leaf, and (**c**) implemented in design software.

**Figure 5 materials-12-01060-f005:**
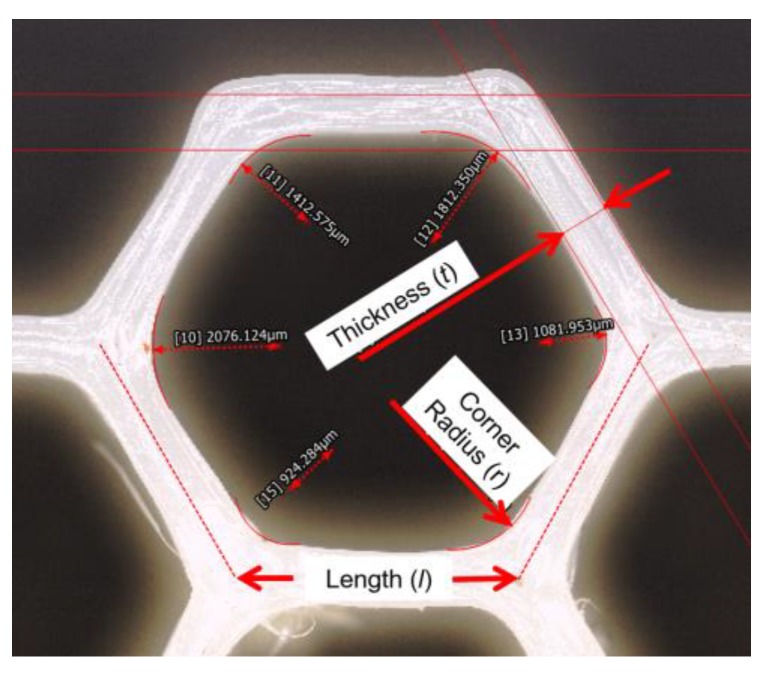
The hexagonal honeycomb can be described in terms of the lengths of its walls (*l*), their thicknesses (*t*), and further, the radius at the corner (*r*).

**Figure 6 materials-12-01060-f006:**
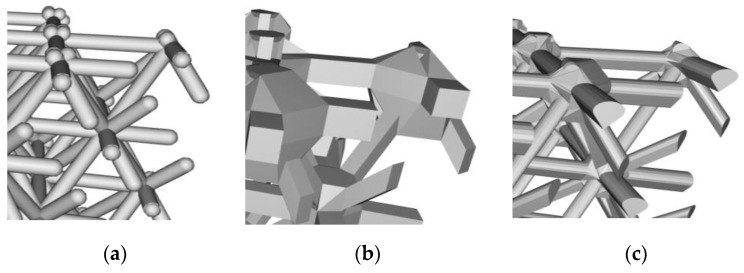
Different cross-sections for the beams that constitute the lattice: (**a**) circular section, (**b**) square section, and (**c**) teardrop shape to aid in self-supporting overhangs.

**Figure 7 materials-12-01060-f007:**
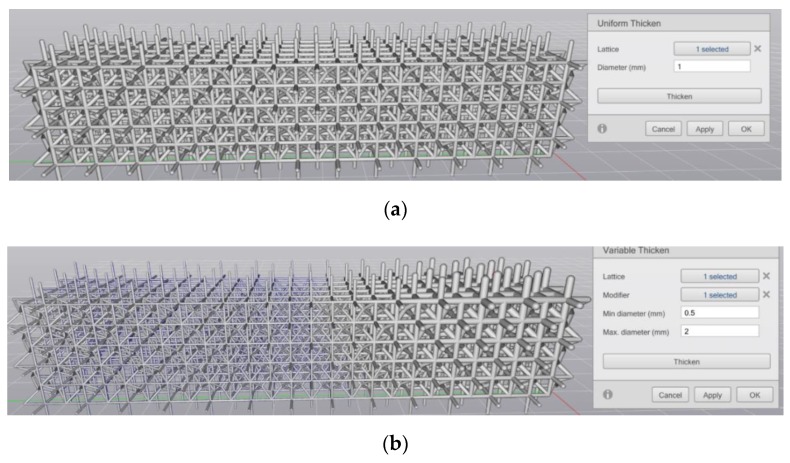
Three methods of prescribing cell thickness: (**a**) global prescription of a thickness value, (**b**) thickness specified in terms of a function, and (**c**) thickness optimized by the solver in response to local stresses (screen captures using nTopology software [9]).

**Figure 8 materials-12-01060-f008:**
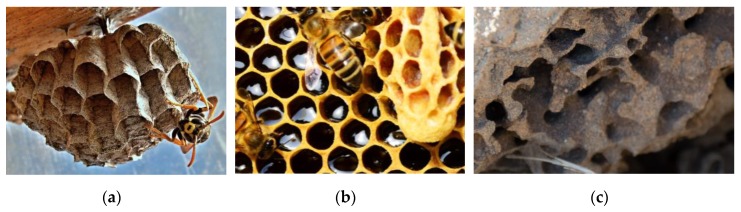
(**a**) Polistes wasp’s nest (made of paper), (**b**) bee’s nest (made of beeswax) and (**c**) termite nest (made of mud) (Attr: P. Asman and J. Lenoble, creative commons 2.0 Generic (CC BY 2.0)).

**Figure 9 materials-12-01060-f009:**
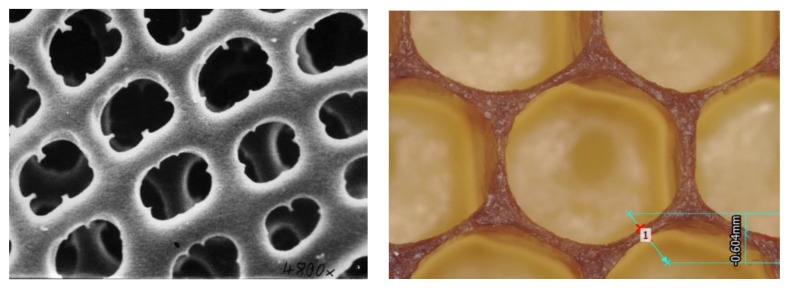
High magnification images of cells comprising a radiolaria (**left**, attr: Hannes Grobe, Wikimedia Commons) and honeycomb (**right**). Clearly visible are nuances such as the corner radius and variations in the beam thicknesses.

**Figure 10 materials-12-01060-f010:**
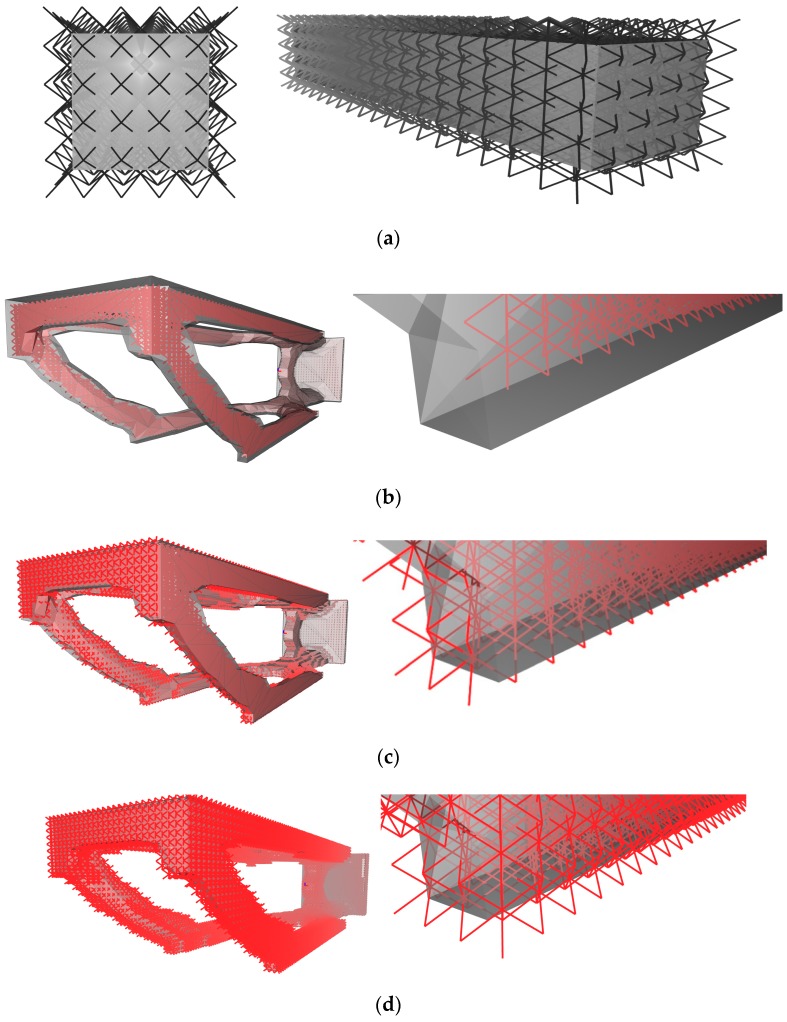
Uniform lattice infill: (**a**) basic concept, (**b**) uniform infill with every unit cell completely represented inside the boundary, (**c**) aligning of the centroid of the cells with the boundary, and (**d**) ensuring the entire structure is completely filled with cells.

**Figure 11 materials-12-01060-f011:**
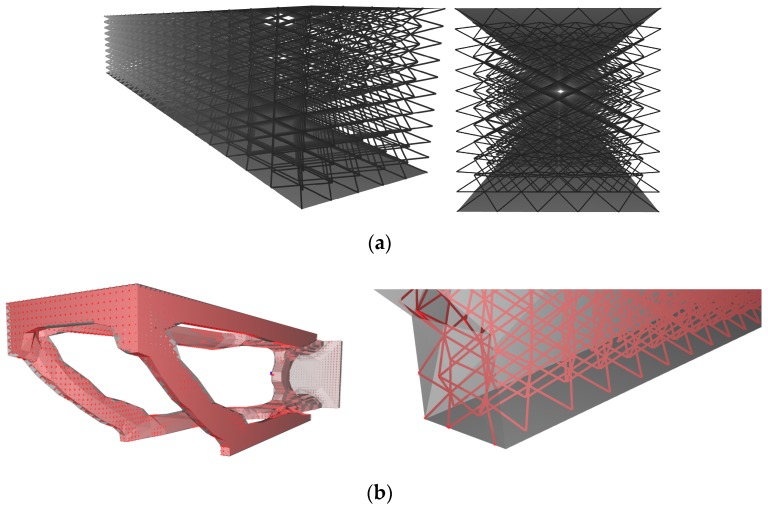
(**a**) Basic conformal lattice infill, and (**b**) wrapping cells such that they conform to the boundary.

**Figure 12 materials-12-01060-f012:**
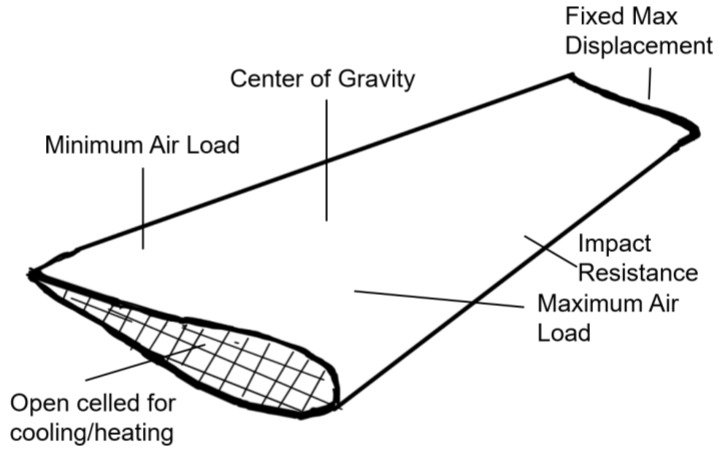
Multi-functional wing concept made out of a cellular material that is both locally and globally optimized for structural and thermal requirements, adapted from reference [6].

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
