# Peer review of "Four Questions in Cellular Material Design"

_materials, 2019, doi:10.3390/ma12071060_

Round 1
Reviewer 1 Report
The manuscript entitled "Four Questions in Cellular Materials Design" (manuscript ID: materials-468371) by Dhruv Bhate provides an overview towards designing the cellular materials based on modeling. Specifically, emphasis is given to the issues associated with development of the cellular materials. This research field is promising for developing new Additive Manufacturing (AM) based materials.
However, in the present form, the manuscript is not recommended for publication as it requires overall improvement in most of the sections. Specifically, a strong motivation for cellular materials in the Sec. 1 is missing as this is the center point of the manuscript. Since AM technique plays important role for designing cellular materials, a description should be included in that aspect.
In most of the sections (Sec. 2, 3 and 5), it looks like reporting the literature only. The discussions related to these should be improved.
Author Response
The author thanks the reviewer for their inputs. Here are responses to each of the points raised by the reviewer, we hope these are appropriate and adequate.
Point 1:
...in the present form, the manuscript is not recommended for publication as it requires overall improvement in most of the sections.
Response 1:
The author wishes to place this publication in context - this paper is being submitted as a "Perspective" submission, and not a conventional research or review. The need for this article was motivated by discussion with several researchers and practitioners in the field of cellular materials, where it emerged that we lacked a framework for discussing the proper methods to design these materials. The intent of this publication is thus not to provide new research results, or provide a comprehensive review, but to provide a perspective on how these 4 design questions can help formulate the design approach for cellular materials.
Point 2:
Specifically, a strong motivation for cellular materials in the Sec. 1 is missing as this is the center point of the manuscript.
Response 2:
The author felt it fit to not discuss the motivation for cellular materials in too much detail since it is considered to be quite a mature area, evidenced by the three textbooks in this field. However, the reviewer is right in pointing out that this discussion is too brief. To correct this, a line has been added to discuss applications of cellular materials.
Point 3:
Since AM technique plays important role for designing cellular materials, a description should be included in that aspect.
Response 3:
A line is added to section 1 to discuss the way in which AM has now enabled the designer to manufacture a range of shapes that were hitherto not possible.
Point 4:
In most of the sections (Sec. 2, 3 and 5), it looks like reporting the literature only. The discussions related to these should be improved.
Response 4:
To keep the Perspective reasonably short, the author has chosen to point the reader towards references where they are available, instead of revisit or discuss them. Nonetheless, the reviewer is correct in implying that that sections 2, 3 and 5 are limited in content in comparison to section 4. This is mostly due to the fact that Section 4 deals with the underlying question of parameter design, which is the area with the most possible variety in design options of all 4 questions discussed here. Nonetheless, several lines have been added to further substantiate the discussion for the other questions and Section 4 has also been edited down by a few lines.
Thank you for your time with this review.
Sincerely,
Dhruv
Reviewer 2 Report
“In the following discussion, we examine each of these questions”
“we identify the key challenges in addressing these questions, with the hope of”
In formal academic writing, third person should be used. Please modify.
Figure 6,
Please using other colors and make them easier to understand, especially image (b)
Equations
Please add references as needed
Conclusions
Please give a brief summary of this study by listing the major findings, which should be separately from the discussion section.
“Cells can be distributed in a prescriptive manner, shown for a periodic lattice (a) or by use of a function, shown for a stochastic lattice (b)”
By using
“optimization based methods for lattice generation also tend to focus on parameter, and not cell size”
“Parameters” would be better here
“The previous two sections dealt with selection of a unit cell topology and its size distribution”
With the section
Author Response
Point 1:
“In the following discussion, we examine each of these questions” “we identify the key challenges in addressing these questions, with the hope of” In formal academic writing, third person should be used. Please modify.
Response 1: All references to first person are now removed and replaced with third person writing.
Point 2: Figure 6, Please using other colors and make them easier to understand, especially image (b)
Response 2: While I was not able to change colors (these are non-editable in the software, I cropped the images to make them clearer)
Point 3: Equations - Please add references as needed.
Response 3: References have been added
Point 4: Conclusions - Please give a brief summary of this study by listing the major findings, which should be separately from the discussion section.
Response 4: I chose not to include conclusions since this is a Perspective (and not original research per se) with no major findings that emerged from this work itself. As a result, it felt a bit presumptuous to be claiming them. I hope that is acceptable.
Point 5: “Cells can be distributed in a prescriptive manner, shown for a periodic lattice (a) or by use of a function, shown for a stochastic lattice (b)” By using
Response 5: Replacement made
Point 6: “optimization based methods for lattice generation also tend to focus on parameter, and not cell size” “Parameters” would be better here
Response 6: Correction made
Point 7: “The previous two sections dealt with selection of a unit cell topology and its size distribution” With the section
Response 7: Actually, that is correct: the sections dealt “with the selection” – no changes made.
Reviewer 3 Report
Indeed, the additive manufacturing technologies combined with the latest software tools for topology optimization and CAD automation have enabled an extensive potential for using cellular structures within structural components.
The author of the article gives an interesting and pertinent view of the current challenges when using cellular structures for designing, trough topology optimization and not only, high performance lightweight components.
Author Response
Thank you for your feedback.